# Critical success factors for high routine immunisation performance: a qualitative analysis of interviews and focus groups from Nepal, Senegal, and Zambia

Zoe Sakas ,[1] Kyra A Hester,[1] Anna Ellis,[1] Emily A Ogutu,[1] Katie Rodriguez,[1] Robert Bednarczyk,[1,2] Sameer Dixit,[3] William Kilembe,[4] Moussa Sarr ,[5] Matthew C Freeman [1]

For numbered affiliations see end of article.

**Correspondence to**
Matthew C Freeman;
matthew.freeman@emory.edu

## ABSTRACT

**Objectives** Vaccination averts an estimated 2–3 million deaths annually. Although vaccine coverage improvements across Africa and South Asia have remained below global targets, several countries have outperformed their peers with significant increases in coverage. The objective of this study was to examine these countries' vaccination programmes and to identify and describe critical success factors that may have supported these improvements.

**Design** Multiple case study design using qualitative research methods.

**Setting** Three countries with high routine immunisation rates: Nepal, Senegal, and Zambia.

**Participants** We conducted 207 key informant interviews and 71 focus group discussions with a total of 678 participants. Participants were recruited from all levels, including government officials, health facility staff, frontline workers, community health workers, and parents. Participants were recruited from both urban and rural districts in Nepal, Senegal, and Zambia.

**Results** Our data revealed that the critical success factors for vaccination programmes relied on the cultural, historical, and statutory context in which the interventions were delivered. In Nepal, Senegal, and Zambia, high immunisation coverage was driven by (1) strong governance structures and healthy policy environments; (2) adjacent successes in health system strengthening; (3) government-led community engagement initiatives, and (4) adaptation considering contextual factors at all levels of the health system.

**Conclusions** Throughout this project, our analysis returned to the importance of defining and understanding the context, governance, financing, and health systems within a country, rather than focusing on any one intervention. This paper augments findings from existing literature by highlighting how contextual factors impact implementation decisions that have led to improvements in childhood vaccine delivery. Findings from this research may help identify transferable lessons and support actionable recommendations to improve national immunisation coverage in other settings.

## STRENGTHS AND LIMITATIONS OF THIS STUDY

⇒ This study was conducted by a multidisciplinary team with authors and advisors from a variety of sectors, including immunisation, epidemiology, political science, implementation science and behavioural science.

⇒ Our team ensured the involvement of stakeholders, researchers and policy makers in the global immunisation sector during the design, implementation and dissemination of this project.

⇒ Through extensive qualitative data collection, we were able to gain an insight from stakeholders at all levels of the vaccine delivery systems in our target countries—from mothers to community health workers, to national-level leadership.

⇒ It was difficult to determine causation because we focused on countries that were successful in vaccine delivery but were unable to carry out a similar analysis in countries with lower vaccination coverage for comparison.

⇒ Using qualitative methods to understand historical events was challenging; interviewees often spoke about current experiences rather than discussing historical factors.

## INTRODUCTION

Vaccination is recognised as one of the most influential public health interventions, averting an estimated 2–3 million deaths annually.[1–4] From 2012 to 2020, the Global Vaccine Action Plan (GVAP) aimed to reach 90% country-level coverage and 80% district-level coverage of the third dose of diphtheria, tetanus, pertussis vaccine (DTP3) among 1-year-old children,[5] a globally recognised proxy for vaccination system performance.[6] However, by 2018, only 95 of the 193 WHO Member States achieved country-level GVAP targets, and less than one-third met

district-level targets.[7–9] In the late 2000s, DTP3 coverage in Africa and South Asia was particularly low, at 74% (2019) and 78% (2016), respectively.[8] However, several countries outperformed their peers with significant increases in routine immunisation coverage from 2000 to 2020.[10] Examining the vaccination programmes of these exemplary countries provides an opportunity to identify and describe critical success factors that may have supported these improvements.

The essential components of an effective vaccine delivery system are well established and include strong governance and leadership, healthcare financing, human resources, a robust supply chain, community engagement and information systems for evidence-based decision-making.[11–13] Research on behaviours related to routine immunisation focuses on recognised determinants of coverage, including intent to vaccinate, community access and health facility readiness.[14] Although existing literature describes vaccine delivery systems, there is a gap in knowledge related to how programmes and policies are implemented, how strategies are operationalised and how context influences programming. By closely examining the vaccine delivery systems of three countries with high vaccination rates, we can draw conclusions about *how* and *why* vaccine programmes work in specific contexts, and how they may be tailored for success elsewhere. This paper augments findings from existing literature by highlighting how structural and contextual factors—including governance, collaboration, the environment and cultural norms—impact implementation decisions that have led to improvements in childhood vaccine delivery.

The purpose of this project was to identify critical success factors that contributed to catalytic growth in childhood routine immunisation coverage in three countries, and to describe how countries with high vaccination rates took similar paths to success.[15–19] We retrospectively analysed improvements in vaccine coverage from 2000 to 2020. The COVID-19 pandemic sent shocks through vaccination programmes worldwide, negatively impacting vaccination coverage. However, the global impact of the pandemic was not examined as part of this study due to our retrospective approach. Findings from this research may identify transferable lessons and support actionable recommendations to improve national immunisation coverage in other settings.[20]

## METHODS

This multiple case-study analysis was conducted using data from the Exemplars in Vaccine Delivery project within the Exemplars in Global Health programme, funded by the Gates Foundation and executed through Gates Ventures.[10 20 21] The Exemplars in Vaccine Delivery project involved qualitative examination of the national vaccine delivery systems of three low-income and middle-income countries—namely Nepal, Senegal and Zambia—with high childhood routine immunisation coverage, compared with their peers.

This paper compares the findings from three individual country-level case studies. We identify and describe factors that impacted successful vaccine delivery in Nepal, Senegal and Zambia while addressing differences based on contextual influence. The individual case-study papers provide additional details regarding country-specific implementation.[15–17]

We used qualitative analysis to investigate factors that contributed to high vaccination coverage through 207 key informant interviews (KIIs) and 71 focus group discussions (FGDs) at the national, regional, district, health facility and community levels. We triangulated these findings with quantitative analyses using publicly available data, which are published elsewhere and referenced throughout this paper.[10 21]

Prior to data collection, we developed a conceptual framework (figure 1) to organise factors that impact childhood vaccine coverage globally. This framework was based on the work of Phillips *et al* and LaFond *et al* alongside a broader review of the vaccine literature.[14 22] Details regarding development are available in our protocol.[10]

### Study sites

Countries were selected based on DTP1 and DTP3 coverage estimates because DTP1 is commonly used as a proxy of access, and DTP3 as a proxy of continued utilisation of immunisation services (online supplemental appendix 1).[10 23 24] Nepal, Senegal and Zambia were selected as exemplary countries due to relatively high DTP1 and DTP3 coverage during the timeframe assessed in this study. Three regions within each country were identified in consultation with national stakeholders and available data (table 1). Partner organisations and Ministry of Health (MoH) (Note that each country has a slightly different name for their MoH (e.g., Ministry of Health and Population in Nepal; Ministry of Health and Social Action in Senegal), but all will be referred to as 'MoH' throughout for simplicity.) officials facilitated site selection and data collection activities. Additional methods for country, region and district selection are provided in our protocol and case studies.[10 15–17]

### Qualitative data collection and analysis

Qualitative data were collected between October 2019 and April 2021 in Nepal, Senegal, and Zambia (table 1). KIIs were held with government officials to understand their current and historical perspectives on the national vaccination programme based on their official positions. Given inherent power dynamics, we decided that FGDs would have been inappropriate. At the community level, we conducted FGDs because we hoped that group conversations would allow for interactive discourse on the key themes. Interview guides were informed by the Consolidated Framework for Implementation Research (CFIR)[25] and the Context and Implementation of Complex Interventions (CICI) framework.[26] KII and FGD guides were translated into local languages by research assistants. All interview guides were piloted before use and adjusted

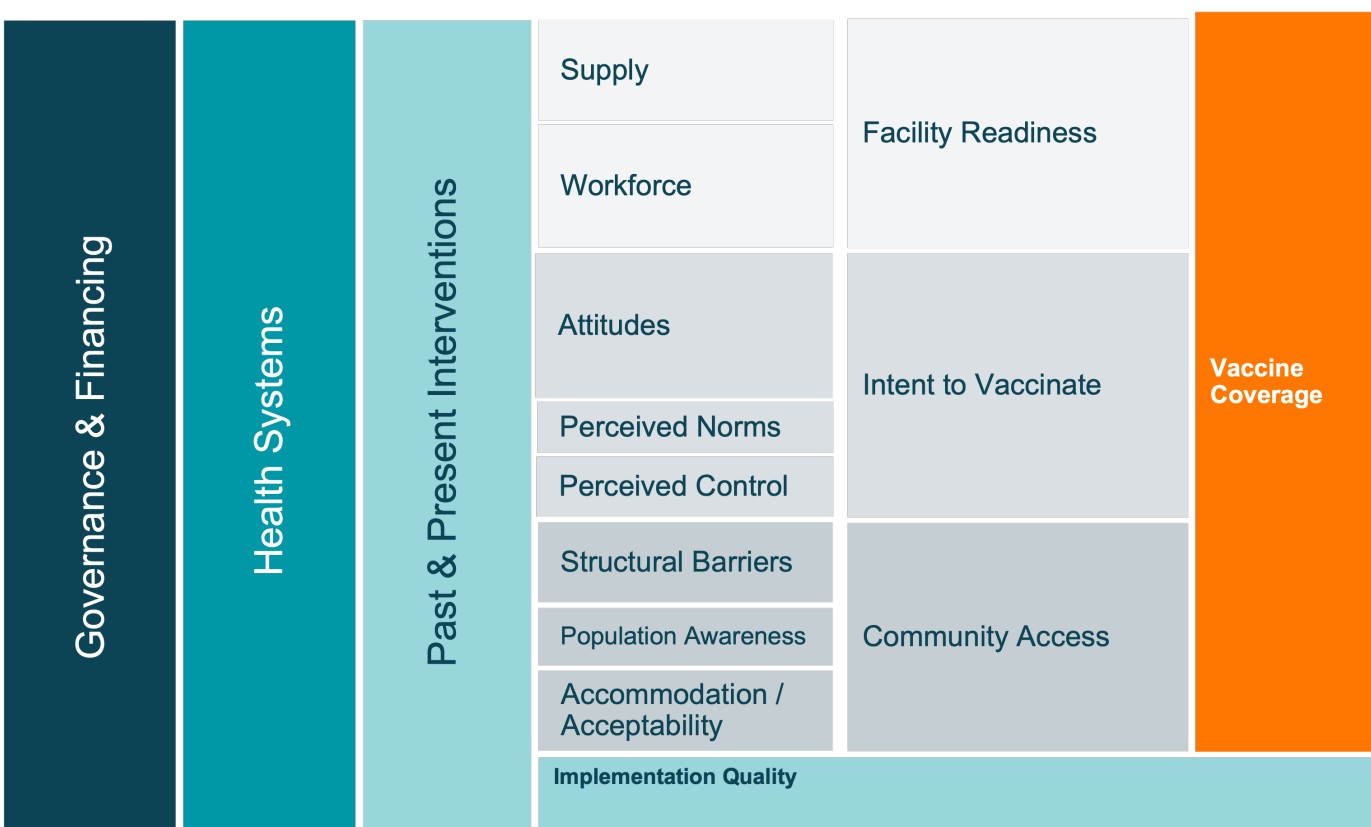

**Figure 1** A priori conceptual framework of the drivers of vaccine coverage.

iteratively throughout data collection. For each country, an initial list of KIIs was developed with local research partners and MoH officials with snowball sampling used to identify additional key informants. Our sampling approach aimed to include a diverse sample of participants regarding geographic location and demographic qualities. Caregivers and volunteer community health workers were recruited for FGDs from health facility catchment areas with the assistance of local health staff. The duration of KIIs and FGDs averaged one and a half hours. KIIs and FGDs were audio-recorded with the permission of participants. Research files, recordings and transcriptions were deidentified and password protected. Data analysis processes were generally consistent in all three countries examined.

We applied thematic analysis of the transcripts to identify critical success factors. Analysis was informed by the domains outlined in figure 1. Data analysis was initially conducted individually for each case study, with separate members of our research team leading analysis for each country. However, to ensure consistency and coherence, all data analysts were involved in the review, coding and summarisation of data from all three countries. We met frequently to discuss overlap and to compare empirical findings. Transcripts were coded and analysed using MaxQDA2020 software (Berlin, Germany). Coding was completed by researchers at Emory, some of whom also developed and revised the codebook. Coding meetings were held frequently to discuss code use and definitions.

Researchers from in-country teams attended additional meetings to discuss data and to ground-truth findings. We considered setting and participant roles while identifying key points and further contextualised data using historical documents and a literature review. Qualitative data collection and analysis tools, including topic guides and codebooks, can be found in online supplemental appendix 2 and on our Open Sciences Framework webpage.[27]

Following analysis of individual case studies, we explored and discussed the similarities and differences between our findings. Using a multiple case-study approach, we developed conceptual ideas that pulled from all three countries to highlight the key components required for successful vaccine delivery (as per analysis of factors from three exemplary countries). Our discussions were rooted in the empirical data and we supported all conclusions with informative quotations and data summaries. When there were discrepancies, we revisited the raw data from individual case studies.

### Supplemental quantitative data analysis
Quantitative data were collected through secondary data sets, which included information from the MOH in Zambia, Nepal and Senegal among other partners. Data were used to estimate routine immunisation coverage from 2000 to 2019 and to uncover trends related to significant improvements. Additional analyses were conducted to identify indicators that may be associated with immunisation coverage success among low-income

**Table 1** Summary of countries, regions, districts selected for research and data collection activities

| Country | Nepal | | Senegal | | Zambia | |
|---|---|---|---|---|---|---|
| In country research partner | Centre for Molecular Dynamics, Nepal | | Institut de Recherche en Santé de Surveillance, de Surveillance Epidémiologique et de Formation (IRESSEF) | | Centre for Family Health Research in Zambia | |
| Data collection period (MM/YYYY) | 8/2019–12/2019 | | 12/2020–4/2021 | | 10/2019–02/2020 | |
| **Regions and districts** | | | | | | |
| Region 1 | Madhes* | Dhanusha, Bara, Mahottari | Ziguinchor | Ziguinchor, Oussouye, Diouloulou | Lusaka | Lusaka, Rufunsa, Chongwe |
| Region 2 | Bagmati | Makwanpur, Dolakha, Kathmandu | Dakar | Rufisque, Mbao, Keur Massar | Central | Chibombo, Chitambo, Serenje |
| Region 3 | Gandaki Pradesh | Kaski, Myagdi, Nawalparasi | Tambacounda | Tambacounda, Koumpentoum, Goudiry | Luapula | Chipili, Nchelenge, Samfya |
| **Key informant interviews** *Key: number of KIIs (participants)* | **79 (79)** | | **62 (63)** | | **66 (85)** | |
| National-level government staff | 11 (11) | | 5 (5) | | 11 (12) | |
| Partner organisation staff | 8 (8) | | 4 (4) | | 11 (15) | |
| Regional health staff | 5 (5) | | 7 (7) | | 6 (8) | |
| District health staff | 15 (15) | | 38 (38) | | 10 (19) | |
| Health facility staff | 23 (23) | | 6 (6) | | 7 (10) | |
| Community leaders | 15 (15) | | 2 (2) | | 10 (10) | |
| Community health workers† | 2 (2) | | – | | 11 (11) | |
| **Focus group discussions** *Key: number of FGDs (participants)* | **30 (191)** | | **19 (128)** | | **22 (132)** | |
| Community health workers† | 9 (60) | | 10 (65) | | 10 (60) | |
| Mothers | 9 (60) | | 9 (63) | | 8 (48) | |
| Fathers | 6 (36) | | – | | 1 (6) | |
| Grandparents | 6 (35) | | – | | 3 (18) | |
| **Total (per country)** | **109 (270)** | | **81 (191)** | | **88 (217)** | |
| **Total (across countries)** | | | | | **278 (678)** | |

*Madhes was referred to as Province 2 prior to 2022 (including during data collection)
†Includes volunteer community health workers, female community health volunteers (FCHV)—Nepal; vaccinators—Nepal; bajenu gox—Senegal; and neighbourhood health committee members—Zambia.
FGDs, focus group discussions; KIIs, key informant interviews.

and lower-middle-income countries using regression frameworks to statistically test financial, development, demographic and other country-level indicators. These quantitative analyses were published elsewhere.[28 29] Findings from the quantitative analyses provided additional context for the qualitative findings which informed the results presented here.

### Patient and public involvement

Throughout the Vaccine Exemplars project—which included individual case studies and the pooled analysis presented in this paper—the study team ensured the involvement of stakeholders, researchers and policy makers in the global immunisation sector during the design, implementation and dissemination of this project. We utilised our Technical Advisory Group (TAG) to develop an initial set of questions for scoping visits to our exemplar countries, where study members met with in-country partners and experts to further refine the direction and nature of the questions. Scoping visit findings were used to create the final data collection tools.[10] As this is a hypothesis generating study, and no intervention was provided, data collection was targeted to conform to general standards of burden. We adhered to standard best-practices for qualitative research, iteratively implementing feedback from experts and the TAG. While there was community involvement (FGDs), the primary focus of data collection was centred on historical stakeholders who played key roles in the immunisation programme.[10] On completion of each country case study, findings were disseminated to in-country

experts and country governments, along with the TAG, allowing for the study team to ground-truth results. The research team at Emory University cohosted regular meetings with the in-country research partners, experts in a variety of relevant fields, government officials and other advisors to ask for feedback and contextualisation. Researchers would present on methods, findings and implications following an engaging discussion with advisors that would lead to improvements in our data analysis processes or presentation of findings. Revisions would be distributed through follow-up email conversations and subsequent phone calls with select advisors. Reports, and manuscripts, of findings were sent to those involved. Feedback from stakeholders was incorporated into this multiple case-study analysis.

More information about how this study addresses local research and policy priorities is included in our Reflexivity Statement (online supplemental appendix 3).

### Sex and gender considerations

Although we did not provide an intervention, the research team took gender differences into consideration. For our FGDs, we recruited both mothers and fathers. FGDs were separated by gender so that participants would feel comfortable sharing their thoughts.

Due to the historical nature of this study, our KII sample of key stakeholders and leaders leaned male due to prior gender biases and the tendency to place men in positions of importance or power. However, present-day positions were often filled by women, and we sampled women for KIIs as appropriate.

## RESULTS

### Drivers of successful vaccination programmes in exemplar countries

Following analysis of data from Nepal, Senegal, and Zambia, we revisited the conceptual framework developed during study conception (figure 1). Although existing literature elaborates on determinants of vaccination coverage (eg, intent to vaccinate, facility readiness and community access), there is a gap in knowledge related to *how* policies and programmes are operationalised to address these determinants—which relies heavily on the (1) existing governance and health systems, (2) engagement of community members (or end-users) and (3) ability to adapt to context to improve implementation.

We revised the initial conceptual framework (figure 1) to more accurately reflect findings from empirical data and to highlight emergent success factors (figure 2). Definitions for the key domains presented in the revised framework can be found in table 2.

### Operationalisation of successful vaccine delivery programmes

In Nepal, Senegal, and Zambia, high immunisation coverage was driven by (1) strong governance structures and healthy policy environments; (2) adjacent successes in health system strengthening; (3) government-led community engagement initiatives; and (4) adaptation of programming based on contextual factors at all levels of the health system.

In Nepal, public codification of health (and therefore, vaccines) as a human right ensured the stability of vaccination programming, increased morale among health workers and helped to generate demand for vaccines.

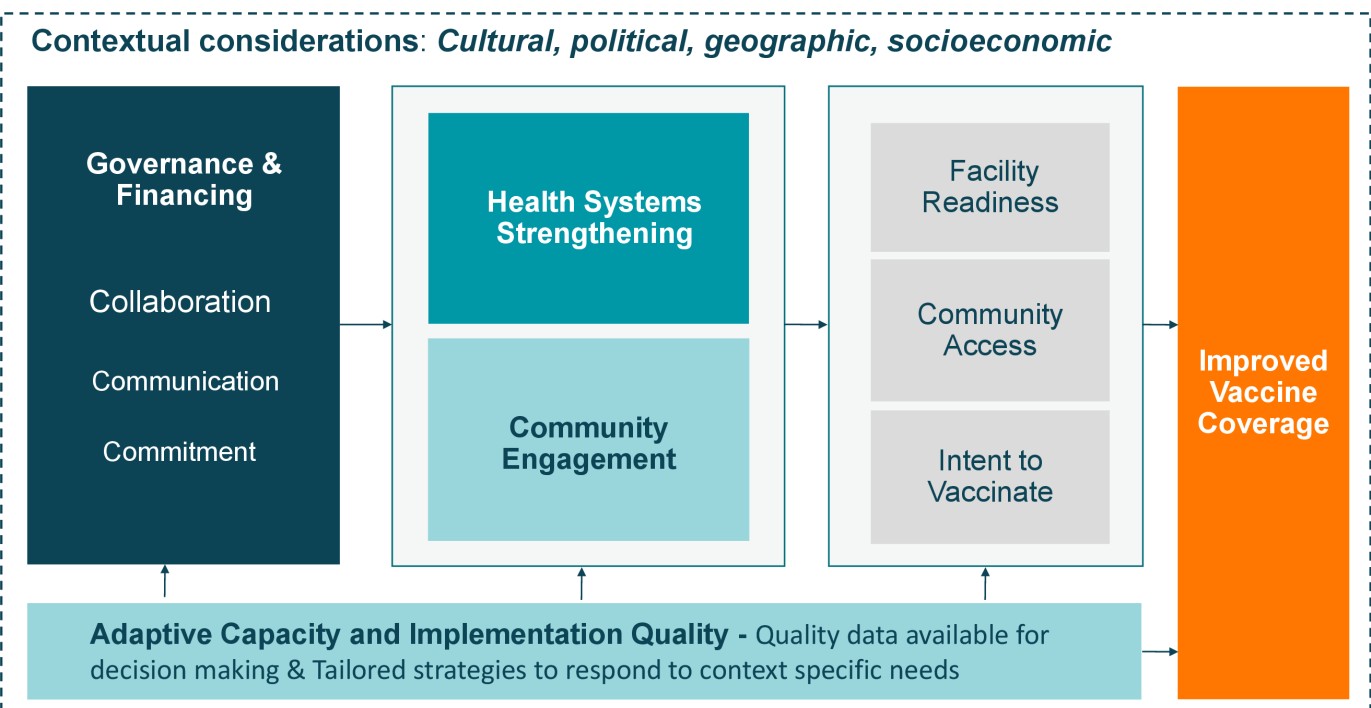

**Figure 2** Revised conceptual framework for drivers of vaccine coverage.

**Table 2** Definitions for drivers of vaccine coverage

| Domain | Definition |
| --- | --- |
| Governance and financing* | The processes that guide policies and operations within the health system, including collaboration (effective working relationships, including alignment of priorities), communication (open and frequent channels for sharing knowledge and feedback) and commitment (dedication to activities and responsibilities). |
| Health system strengthening* | Strategies that are designed to sustainably improve health system performance (across all sectors) through an integrated and horizontal approach, with a focus on equity. |
| Community engagement* | Responsibility of the community to implement vaccination programme activities through local leadership and volunteers, fostered by government initiatives and cultural values. |
| Adaptive capacity* | Capability to sufficiently consider contextual factors, supported by timely evaluation of quality data, to tailor vaccine programming based on local needs and priorities—and improve implementation quality. |
| Intent to vaccinate† | Demand for childhood vaccines on the part of the caregiver or guardian who would result in vaccination in the absence of other barriers. |
| Facility readiness† | Supply (by the health system) of health and vaccination services to adequately meet demand. Incorporates supplies (vials, syringes, etc.), human resources and the consistency of their availability. |
| Community access† | The ability (or inability) to successfully carry out the transaction of vaccine utilisation—for example, a community member's ability to access vaccines should they decide to get vaccinated—that is, barriers and facilitators between intent and readiness. |

*Developed from our research, both via data analysis and desk review of existing literature.
†See figure 1 (and reference Phillips *et al* paper).

Female community health volunteers (FCHVs) and other community health workers lead a powerful community-based approach to demand generation, health education and vaccine delivery, which was supported by cultural values including collective responsibility and community ownership of vaccine coverage.[16] In Senegal, a strong political will supported the prioritisation of vaccination programming, including urgent allocation of resources and cold chain improvements. Long-term partnerships between government agencies and external partners allowed for innovation, capacity building and efficiency within the health system. Finally, improved surveillance efforts led to timely and evidence-based decision-making.[17] In Zambia, effective collaboration was essential to successful vaccine delivery. The Inter-Agency Coordinating Committee (ICC) led long-term collaboration efforts at the national level and supported evidence-based programming through multiple technical working groups. At the local level, Neighbourhood Health Committees spearheaded community-driven strategies via community action planning and strengthened the link between communities and health facilities.[15]

Below, we summarise similarities apparent in all three high-performing countries. More information on how policies and programmes were implemented and adapted to local context is available elsewhere.[15–17] Informative quotes from each country and domain can be found in online supplemental appendix 4.

### Strong governance structures and a healthy policy environment

Effective collaboration, open communication and commitment to health and vaccines supported policy development and implementation of the vaccination programmes in Nepal, Senegal, and Zambia.

Collaboration and communication enabled alignment of priorities, division of responsibilities and coordination between levels. In all three countries, collaboration was apparent between external partners and the government, and between different levels of the health system. National fora were established for frequent and multidisciplinary communication, including ICC.[30] Regional, district and community level reflection meetings were held to share and review data and gather feedback for strategic planning. Formal channels for supportive supervision and feedback, both top-down and bottom-up, supported service delivery improvements and reporting.

In all three countries, vaccination was a 'priority' or 'flagship" programme recognised by the MOH. Prioritisation of vaccination within the health sector supported sufficient resource allocation. In Nepal and Senegal, dedication to the health sector more broadly was codified through defining health as a right in their national constitutions. Commitment to health and vaccines underpinned policy development and enabled consistent service delivery. Community members could expect reliable and equitable access to vaccines due to the prioritisation of activities by health personnel and the dedication of volunteer community health workers who supported outreach activities and spearheaded demand generation and educational campaigns.

A provision was made in the constitution, and (vaccines) became a fundamental right, so the government has a responsibility to provide. And

management of currently running programs also (focuses on) children who have missed vaccines. (Nepal Health Research Council, Nepal)

## Adjacent successes in health system strengthening

Improvements in the broader health systems of Nepal, Senegal and Zambia—including health postexpansion, training of volunteer community health workers, general capacity building and prioritisation of equitable health services—supported high immunisation coverage.

Health postexpansion and capacity building at subnational and community levels were mentioned by most key informants in all three countries as essential to improvements in vaccination coverage. Building health facilities in rural areas especially allowed parents to easily vaccinate their children, without having to travel far distances or rely on outreach. Improvements in vaccine service delivery also relied on improvements in general infrastructure including roads, electricity and connectivity required for transportation, health facility operations and data reporting. In all three countries, general health system strengthening was centred around equity as a priority for decision-making. Health postexpansion and community health worker programmes were strengthened to prioritise service delivery in hard-to-reach communities, minority populations and remote areas.

> Previously, there was no public health department at a grassroots level, it was just at the National level… So, now the Public Health Specialist is on the ground to try to see that this primary health care activity is promoted. And I think with that supervision, with the introduction of the public health nurses, Health promotion departments, I think these interventions couldn't be achieved higher in terms of immunization activities. (District Health Director, Zambia)

Ongoing efforts to address equity concerns include: (1) overcoming geographical barriers through health postexpansion and outreach services supported by national-level policy and external partner funding; and (2) addressing social, economic and cultural factors through tailored messaging and local ownership of vaccination programming.

Integrated decision-making across health and social sectors, through a horizontal rather than vertical approach, facilitated effective allocation of resources (eg, monetary, human, technical and educational) could be efficiently allocated across programmes. For example, national fora and committees in all three countries included representatives from the immunisation sector, maternal and child health, education and development and financial offices. During meetings, sectors would align long-term priorities and discuss how to reach overall goals in terms of health and social outcomes.

## Government-led community engagement initiatives

Community ownership of vaccination activities and corresponding health outcomes was apparent in Nepal, Senegal and Zambia, and even more so in high-performing districts within these countries. Responsibility of the community to implement vaccination programming through local leadership and volunteers was fostered by government initiatives and cultural values. National policies focused on involvement of traditional leaders and all national-level stakeholders reported on the importance of community actors. In some communities, female volunteers would work beyond their mandates to ensure that all children were vaccinated—a decision motivated by dedication to their communities at large, empathy for the children in their neighbourhoods and a vague understanding of herd immunity.

Key informants from the national to community level reported that demand for vaccines was critical for improving routine immunisation coverage. In addition to the work of volunteer community health workers, demand generation and public awareness were accomplished through national policies, media engagement, involvement of community actors and school outreach activities.

> Community health workers play a very important role—they are in direct contact with the population. They are the ones who live in the community, and they identify more with the community. So, there is a relationship of trust between them and the population. (National-level government stakeholder, Senegal)

Further analysis of community health worker programmes and demand generation activities in Nepal, Senegal, and Zambia will be detailed in our upcoming publications.[31]

## Adaptation at all levels of the health system

In Nepal, Senegal and Zambia, decision-makers and implementers at the national, subnational, and community levels demonstrated their ability to consider contextual factors, evaluate or reflect on quality data and tailor vaccine programming based on local needs and priorities.

Quality data were available for evidence-based decision-making in all three countries. Data software was implemented at the district level in all countries—or in the case of Senegal, at the health postlevel—allowing stakeholders at all levels to utilise accurate and up-to-date coverage estimates. Lack of quality data in some areas may have hindered this process. However, when data were not available, community actors would adapt programming based on head counts conducted by volunteers and an anecdotal understanding of the contextual and cultural needs in their communities.

> We have been advising facilities that they need to own the data; they have to make sure that they utilize it by analyzing… If they are not meeting their targets, they need to reflect on the data and see where they are going wrong… For those (health facilities) that are performing well, we find out what they are doing,

**Table 3** Key interventions and programmes implemented to improve routine immunisation coverage in Nepal, Senegal and Zambia from 2000 to 2019, based on KII and FGD data

| Category | Policy or intervention | Zambia | Nepal | Senegal |
|---|---|---|---|---|
| National-level programming Includes external partners and government | Introduction of new vaccines | Blue | Blue | Blue |
| | Vaccination as 'priority' programme | Green | Blue | Green |
| | Strategic reallocation of targeted funds | Blue | Blue | Pink |
| | Health postexpansion | Green | Blue | Blue |
| | Constitution codifying health as a right | Pink | Blue | Green |
| | Forums for decision-making (e.g., ICC) | Blue | Green | Blue |
| | Media engagement | Blue | Blue | Blue |
| Subnational programming Regional and district level | RED/REC implementation | Blue | Green | Blue |
| | Training community health workers | Green | Blue | Blue |
| | Meetings for evidence-based decisions | Blue | Blue | Green |
| | Data software, health post level | Pink | Pink | Green |
| | Data software, district level | Green | Blue | Blue |
| | Cold chain expansion | Blue | Green | Blue |
| Community programming Health post and village/community level | Community health worker programme | Blue | Blue | Blue |
| | Community-level health committees | Blue | Green | Blue |
| | Outreach services | Blue | Blue | Blue |
| | Microplanning in health facilities | Blue | Green | Green |
| | Engagement of community leaders | Blue | Blue | Blue |
| | Support groups for caregivers | Green | Blue | Blue |
| | School outreach for health education | Green | Green | Green |
| | Promotion through media (e.g., radio) | Blue | Blue | Blue |

Key for reading the table below:
■ Not mentioned by key informants or community stakeholders who participated in this study.
■ Implemented; participants noted that there was room for improvement (eg, gaps or challenges).
■ Implemented; participants described successful implementation (eg, exceeds recommended guidelines).
FGDs, focus group discussions; ICC, Inter-Agency Coordinating Committee; KIIs, key informant interviews.

and we try to replicate to the other facilities that are not doing as well. (District Health Officer, Zambia)

Adapting strategies to respond to context-specific needs was essential for reaching children. In all three countries, flexible vaccine delivery days and outreach services supported unique scheduling needs (eg, in Senegal, vaccinating children in the late afternoon during farming season). Outreach strategies were also tailored to accommodate contextual factors and address barriers to access. Understanding cultural and social values enabled community actors to tailor demand generation activities so that community members would be more apt to adopt positive health behaviours. Some examples of this may be using a local language, involving religious or community leaders in engagement and sensitising, or addressing cultural or social barriers directly through personal communication.

### Interventions and programmes across high-performing countries

Key informants discussed a myriad of interventions and programmes that they believed led to the improved and sustained vaccination coverage in their countries (table 3). Interventions are organised by implementation level; if an intervention was implemented at multiple levels of the health system, it may be repeated in the table. Although the specific components of vaccine programming were not identical across countries, many of the general themes overlapped (eg, the scope of community-level health committees varied between countries, but the purpose/objective of these groups was similar).

### DISCUSSION

Notions of strengthening national governance structures have historically been recognised through country-level application of global guidelines to improve vaccine delivery and health systems; however, existing literature lacks concrete descriptions of what this entails. This paper augments findings from previous research by highlighting how structural and contextual factors—including governance, collaboration, the environment, and cultural norms—impact implementation decisions that have led to improvements in childhood vaccine delivery. We developed evidence-derived operational definitions of structures and processes that we identified as critical to the success of vaccine delivery systems in countries with high

routine immunisation coverage. We described *how* these processes were utilised in high-performing countries to drive catalytic improvements in coverage. Previous research has not explored the details or importance of governance, financing, health systems, or other contextual factors; our original framework based on pre-existing literature (figure 1) was missing complexity in these components, which is indicative of the limited knowledge in this area. In order to understand how global guidelines should be adapted to reflect local priorities and challenges, it was important for us to describe how national governance structures operate with the consideration of contextual factors. Ignoring a structured approach to understanding these factors could lead to repeated instances of suboptimal programme performance.

Our data points to the need for programmers and policy makers to better understand underlying governance structures. The structures and processes we examined in Nepal, Senegal and Zambia supported adaptive decision-making based on context-specific needs and priorities. Adaptation at all levels enabled tailored and targeted programming based on evaluation of high-quality data. While health system improvements led by national governments were essential, our data also highlighted the importance of a deliberate and consistent focus on community engagement, which aligns with existing literature.[1 27 28] Community involvement in vaccine programming facilitated the government's understanding of cultural and social barriers and enablers which contributed to effective adaptation of interventions to increase coverage. Community ownership was fostered through national policies and resources for capacity building, engagement of traditional leaders and the strength of the community health worker programmes.

We found that many improvements in immunisation coverage emerged from adjacent successes in health system strengthening. While an unsurprising finding that health system strengthening would be an essential ingredient in support of vaccine system strengthening, it points to the need for a more integrated, rather than siloed, approach to child health. Health system strengthening has become a priority in recent decades as disruptions due to natural disasters, disease outbreaks or civil unrest have exposed weaknesses in health systems.[32 33] Although we focused on examining success factors for vaccine delivery, our data also suggest that there are still gaps and challenges in exemplar countries, including: (1) equitable access (eg, lack of health posts in rural and remote areas); (2) long-term financing (eg, most funding comes from external partners); (3) poor infrastructure (eg, roads and electricity) and (4) a heavy reliance on volunteer community health workers which experienced vacancies and decreased morale in some cases.

In Nepal, Senegal, and Zambia, overall commitment to health and vaccines strengthened governance processes and supported vaccination efforts through prioritisation of activities and resource allocation. Following the completion of GVAP in 2020, stakeholders reported that lack of consideration of existing structures and contextual factors limited the plan's success.[34] In addition, the monitoring and evaluation processes proposed through GVAP were often seen as unrealistic.[9 34 35] Suggested improvements included engaging with countries to accurately and sufficiently consider local context while setting global goals and targets and promoting country ownership of strategic planning.[34 35] Our research revealed that national government ownership of strategic planning was critical to these countries' success, which aligns with the critical reflection of GVAP and other existing literature.

Overall, the strategies implemented by exemplar countries align with the WHO Comprehensive Framework of Strategies and Practices for Routine Immunisation, which focus on managing the immunisation programme (eg, political commitment and partnerships), mobilising communities, generating demand and monitoring progress.[1] Throughout the study, our analysis returned to the importance of defining and understanding the context, governance, financing and health systems within a country, rather than focusing on any one intervention. This highlights the importance of further research into these structural factors. We examined how these strategies were operationalised to improve coverage in Nepal, Senegal and Zambia, and how they can be applied in other settings. Understanding how governance processes function in high-performing countries may allow others to build stronger systems, adapt guidelines and reflect on progress made in the field. Our research introduces areas of inquiry for further investigation, as the critical success factors that primarily emerged in all three exemplar countries are not the current focus within existing literature around determinants for vaccine coverage.

This study has several limitations. First, it was difficult to determine causation because we focused on countries that were successful in vaccine delivery but were unable to carry out a similar analysis in a country with lower vaccination coverage to compare. Second, the research tools focused on the factors that drove catalytic change and did not focus on interventions or policies that were unsuccessful. Third, using qualitative methods to understand historical events was challenging; interviewees often spoke about current experiences rather than discussing historical factors. However, we probed respondents to reflect on longitudinal changes in the immunisation programme. Additional country-specific limitations can be found elsewhere.[15–17]

## CONCLUSION

Through focusing on countries with high routine immunisation coverage, we examined how vaccine delivery systems may leverage components of existing governance structures and health systems to accelerate and sustain coverage. Operational definitions for governance, health system strengthening, community engagement, and adaptive capacity, along with descriptions of how these processes were implemented in high-performing

countries, may help other countries implement similar improvements. Our findings highlight the need for programme managers and policy makers to understand and consider the strengths and limitations of existing structures while adapting to the emergent needs and priorities relevant to the context. Our study looked retroactively on *how* and *why* countries succeeded in achieving high early childhood immunisation coverage. The COVID-19 pandemic has highlighted the fragility in our health systems, and specifically how robust vaccine systems can rise to the challenge of emergent vaccination needs. While our findings are not immediately applicable to the current COVID-19 vaccination needs, our underlying approach to understand context remain relevant to consider as countries strive to vaccinate their populations in the ongoing pandemic.[36]

**Author affiliations**
[1]Gangarosa Department of Environmental Health, Rollins School of Public Health, Emory University, Atlanta, Georgia, USA
[2]Hubert Department of Global Health, Emory University, Atlanta, Georgia, USA
[3]Center for Molecular Dynamics Nepal, Kathmandu, Nepal
[4]Rwanda Zambia HIV Research Group, Emory University, Lusaka, Zambia
[5]Institut de Recherche en Santé de Surveillance Epidémiologique et de Formations, Dakar, Senegal

**Acknowledgements**  We gratefully acknowledge the participants who gave their time and insights to help us better understand the vaccine delivery systems of Zambia, Nepal and Senegal, along with facilitators from their respective Ministries of Health. Contributions were also made by Sarah Chesemore, Anna Rapp, Tove Ryan and Ethan Wong from the Bill and Melinda Gates Foundation; Kate Buellesbach, Nancy Fullman, Nathaniel Gerthe, Gloria Ikilezi, Caitlyn Mason, David Phillips and Oliver Rothschild, Jordan-Tate Thomas and Angela Wang from Gates Ventures and the Vaccine Exemplars Research Advisory Group, including Agnes Binagwaho, Laura Craw, Carolina Danovaro, Anuradha Gupta, Heidi Larson, Penelope Masumbu, Kate O'Brien, Helen Rees, Lora Shimp and Aaron Wallace. More information about how this study addresses local research and policy priorities is included in our Reflexivity Statement.

**Contributors**  KAH, AE, SD, WK, MS, RB and MCF: project conceptualisation and methodology; KAH, ZS, AE, KR, SD, WK, MS, RB and MCF: investigation and data curation; KAH, ZS, ASE, KR, EAO, RAB and MCF: formal analysis; ZS and KAH: writing—original; KAH, ZS, ASE, EAO, KR, SD, WK, MS, RAB and MCF: writing—review and editing; all authors provided approval of the final version. MCF, as guarantor, is responsible for the overall content

**Funding**  This work was supported by the Bill and Melinda Gates Foundation, Seattle, WA (OPP1195041) with a planning grant from Gates Ventures, LLC, Kirkland, WA.

**Competing interests**  The authors declare that they have no known competing financial interests or personal relationships that could have appeared to influence the research reported in this paper.

**Patient and public involvement**  Patients and/or the public were not involved in the design, or conduct, or reporting or dissemination plans of this research.

**Patient consent for publication**  Not required.

**Ethics approval**  This study involves human participants but this study was considered exempt by the Institutional Review Board committee of Emory University, Atlanta, Georgia, USA (IRB00111474); approved by the Nepal Health Research Council (NHRC; Reg. no. 347/2019) in Kathmandu, Nepal; the National Ethical Committee for Health Research (CERNS; Comité National d'Ethique pour la Recherche en Santé) in Dakar, Senegal (00000174); the University of Zambia Biomedical Research Ethics Committee (Federal Assurance No. FWA00000338, REF. No. 166-2019) and the National Health Research Authority in Zambia. All ethics committees abide by the principles of the Declaration of Helsinki; all participants provided written consent exempted this study. Participants gave informed consent to participate in the study before taking part.

**Provenance and peer review**  Not commissioned; externally peer reviewed.

**Data availability statement**  Data are available upon reasonable request. Data are not publicly available as all data are confidential. Deidentified data may be available upon request.

**ORCID iDs**
Zoe Sakas http://orcid.org/0000-0003-4979-8757
Moussa Sarr http://orcid.org/0000-0003-2372-6632
Matthew C Freeman http://orcid.org/0000-0002-1517-2572

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
