## [Reviewer comments · BMJ Open]

ARTICLE DETAILS

TITLE (PROVISIONAL)	Critical success factors for high routine immunization performance: a qualitative analysis of interviews and focus groups from Nepal, Senegal, and Zambia
AUTHORS	Sakas, Zoe; Hester, Kyra; Ellis, Anna; Ogutu, Emily; Rodriguez, Katie; Bednarczyk, Robert; Dixit, Sameer; Kilembe, William; Sarr, Moussa; Freeman, Matthew

VERSION 1 – REVIEW

REVIEWER	Haq, Zaeem GAVI the Vaccine Alliance, EMRO Constituency
REVIEW RETURNED	07-Apr-2023

GENERAL COMMENTS	It is an important study on a topic of public health significance. The authors have tried to develop a case study of three case studies but have been unable to display clarity of concept and the methodological rigor that was required. Firstly, the authors have not established a clear knowledge gap that currently exists. Building on this, they could lead to reader to the research question/s and study objectives, which is missing. In Methods, it is not clear what is the relationship of the present study with the three case studies. The present study appears to be an amalgamation of case studies from three countries that have already been published as preprint or a peer-reviewed article. Nowhere it tells whether it is going to examine the pooled data from all three, nor it lays out the basis of qualitative triangulation that it might do. Throughout the method section the steps are the same as for the individual case studies published earlier. Even the number and breakdown of study participants from all three countries are the same as in those papers. For this study, the authors need to elevate their methodological considerations from the earlier papers. With the overall confusion about what information gap this paper addresses and the lack of definitions provided on the proposed concepts like “structural elements” and “contextual factors”, the reader feels at a loss while trying to find a research question and study objectives to connect the dots with their counterparts in the Methods, Results and Discussion section. In the current shape this manuscript is not worthy of publication in the BMJ open. Considerable rethinking and overhaul would be
---

	required for a second iteration if the other reviewers recommend that path.
--	---

REVIEWER	Leask, Julie University of Sydney, School of Public Health
-----------------	---

REVIEW RETURNED	10-Apr-2023
-------------

GENERAL COMMENTS	This study examined success factors in three countries that had improved routine immunisation coverage. The methods are extensive and include multiple participant categories, groups and informants in each setting. The study was clearly a very large, complex and is likely to be useful for the countries in it, given their involvement. It will also provide useful information for other countries with weaker immunisation programmes. The findings of the study aren't as clearly connected to the data as qualitative studies often are, and don't usually provide concrete illustrations in a way that makes the themes come alive. This is acknowledged as an inevitability for a study that is so big-picture and the authors point to other papers. However, the authors might consider how to be more specific in certain sections, or to bring more of the quotes and narrative accounts from participants into the results. Introduction – This is very well written and addresses some key trends and globally important indicators and their context. There should be a mention of the further decline in coverage after 2019 related to the pandemic, particularly at line 14. As of 2021, coverage had not plateaued, it had experienced a decline in coverage for multiple antigens. Methods. Explain briefly the Exemplars project. The conceptual model – there is no report of any testing of the model and limited reporting of how it was established. It would be best to call it something other than a model (see later for an alternative suggestion). The APA define models as “a theory, usually including a mechanism for predicting psychological outcomes, intended to explain specific psychological processes.” Figure 1 is assumed to be the boxes, but it's not labelled thus in the submission. Assuming that it is the first box, the concepts in it are comprehensive but some are not clearly distinctive. For example, perceived norms/control and attitudes. Or perceived norms and population awareness. What is accommodation/acceptability? While later a table describes these, they relate to the second figure. To reduce cognitive load for readers, acronyms should be minimised (and removed in the abstract). Rather than KIIs and FGDs, is it possible to call these focus groups and interviews? The authors could begin by explaining they undertook key informant interviews, thereafter calling them interviews, and focus groups, thereafter calling them groups. Focus groups are by their very nature discussions – making this word redundant. While I realise these terms are common in global health, it is at least worth considering.
---

	Study sites – while the authors describe a method for selecting sites elsewhere, this article should at a minimum explain briefly the rationale in the methods. The qualitative analysis is described thoroughly. It was theory informed but which theory/theories were used? Was it the conceptual model or other theories? If it was the model, that is not a theory. The authors should state why interviews were used with key informants and focus groups were used with caregivers and community workers. It's a reasonable approach, but each method has unique qualities and should be justified. For the qualitative analysis, who did the coding, who analysed the codes, and how were other team members involved? Section 2.3 – what are catalytic improvements? Also, the quantitative analyses were published elsewhere but no reference is given. Findings were disseminated to experts and governments which is said to “ground truth” the results. It would be useful to hear about the mechanisms of feedback which enabled this ground truthing to occur. Presumably, based on the reflexivity statement, there was a way for countries to reflect back and refine the interpretations? Furthermore, it would be interesting to see how this affected the results, for example when the author’s emerging analysis was revised due to the feedback. Could this be mentioned in the reflexivity statement? Twice it is said the study was hypothesis generating and not intervening. This is unneeded repetition. Results The authors revise figure 1 because of the over-riding importance of cultural, historical and statutory factors, but the new framework doesn't include these named factors. Openness to a new framework (fig 2) is a sign of good qualitative data analysis. Definitions are given for key domains, which is useful, but that makes figure 1 even more distracting. A suggestion for the authors to consider: in the introduction, state that certain concepts were used to structure what was explored in the data collection, and list these and their sources, but leave figure 2 to be a “conceptual framework” and revise this to include the cultural, historical and statutory factors. In table 2, the definition of community access isn't sufficiently clear and relies heavily on jargon. This is the same with describing an “integrated and horizontal” approach in Health Systems Strengthening. Use plainer language to increase access to the meaning for readers. As the paper repeatedly highlights, each country's context (eg., history and governance structures) is unique and this is important background. Is it possible for authors to briefly describe each country to give readers context. The results need to better explain how contexts relate to each of the main themes as these connections aren't clearly made, despite the importance of context. The results are comprehensive and have face validity. It's a pity to divorce most of the illustrative quotes from the description of the
--	--

	findings since that gives them more authenticity. I prefer to see quotes in context, not in other tables. The results don't speak much about the views from the focus groups. Table 3 – change colours to greyscale for printed versions as blue and green don't differentiate. Discussion. This is a key sentence, but its meaning isn't clear enough: “Describing how national governance structures operate with the consideration of contextual factors was essential to our understanding of how global guidelines should be adapted to reflect local priorities and challenges.” Line 35 – at this point, there is a puzzling absence of any discussion of the impact of the pandemic. In Senegal, this would have been discussed in interviews given the timeframe for data collection. A success of communication was that it was both top down and bottom up. Depending on its importance, the authors could consider further reflecting on this characteristic in the discussion, which has been found elsewhere to create resilience against other risks. In the supplementary materials, the search strategy is about falls. Please clarify.
--	--

VERSION 1 – AUTHOR RESPONSE

7	It is an important study on a topic of public health significance. The authors have tried to develop a case study of three case studies but have been unable to display clarity of concept and the methodological rigor that was required. Firstly, the authors have not established a clear knowledge gap that currently exists. Building on this, they could lead to reader to the research question/s and study objectives, which is missing.	Reviewer 1 Dr. Zaeem Haq, GAVI the Vaccine Alliance, Health Services Academy	We appreciate this feedback. Because this paper is comparing and collating findings from three separate case studies, we may have pared down the language and left context out that is more clearly described in the individual country papers. We have edited the introduction to include more information regarding the purpose of this multiple-case study. Added: Although existing literature describes vaccine delivery systems, there is a gap in knowledge related to how programs and policies are implemented, how strategies are operationalized, and how context influences programming. By	Introduction Page 3; Lines 128 - 135
---	---	--	--	--

			closely examining the vaccine delivery systems of three countries with high vaccination rates, we can draw conclusions about how and why vaccine programs work in specific contexts, and how they may be tailored for success elsewhere. This paper augments findings from existing literature by highlighting how structural and contextual factors impact implementation decisions that have led to improvements in childhood vaccine delivery.	
8	In Methods, it is not clear what is the relationship of the present study with the three case studies. The present study appears to be an amalgamation of case studies from three countries that have already been published as preprint or a peer-reviewed article. Nowhere it tells whether it is going to examine the pooled data from all three, nor it lays out the basis of qualitative triangulation that it might do.	Reviewer 1	Addressed. Our approach was to analyze and highlight the key themes across the case studies. We did analyze pooled data across countries; we believe this provides considerable value, as it may allow other countries to glean key lessons. Added: This paper comprises a multiple-case study, which compares the findings from three individual country-level case studies. This comparison was design to review which success factors were operationalized in all three exemplary countries, as well as highlight how differences in context and resources impacted operationalization of the key strategies that emerged from our qualitative data. Here, we outline the main factors that impacted successful vaccine delivery in Nepal, Senegal, and Zambia; the individual country-level papers, provide additional detail regarding country-specific implementation.	Methods Page 4; Lines 165 - 170
9	Throughout the method section the steps are the same as for the individual case studies published earlier. Even the number and breakdown of study participants from all three countries are the same as in those papers. For this study, the authors need to elevate their methodological	Reviewer 1	We decided to address – and hopefully elevate – the methodology by making significant edits to the qualitative data analysis section (section 2.2) in addition to adding the following paragraph to the Methods section: Following analysis of individual case studies, we explored and	Methods, Section 2.2., Page 5; lines 235 - 240

	considerations from the earlier papers.		discussed the similarities and differences between our findings. Using a multiple-case study approach, we developed conceptual ideas that pulled from all three countries in order to highlight the key components required for successful vaccine delivery (as per analysis of factors from three exemplary countries). Our discussions were rooted in the empirical data, and we supported all conclusions with informative quotations and data summaries. When there were discrepancies, we revisited the raw data from individual case studies as needed.	
1 0	With the overall confusion about what information gap this paper addresses and the lack of definitions provided on the proposed concepts like “structural elements” and “contextual factors”, the reader feels at a loss while trying to find a research question and study objectives to connect the dots with their counterparts in the Methods, Results and Discussion section.	Reviewer 1	This was thoughtful feedback; we feel we sufficiently addressed this comment – along with other comments - by: - Adding a description of the gap in knowledge and objectives of this paper in the Intro - Adding significant information regarding the cross-country analysis of the qualitative data to section 2.2. in the Methods - Defining “structural and contextual factors” throughout the paper	✓ Page 2; Lines 111-114 ✓ Page 5; Lines 22 2 --240 ✓ Page 7; Lines 32 9 - 334
1 1	In the current shape this manuscript is not worthy of publication in the BMJ open. Considerable rethinking and overhaul would be required for a second iteration if the other reviewers recommend that path.	Reviewer 1	We appreciate the thorough and thoughtful review. We have made substantial changes to the manuscript per the feedback from yourself and reviewer #2. We hope you find your concerns addressed.	n/a
1 2	This study examined success factors in three countries that had improved routine immunisation coverage . The methods are extensive and include multiple participant categories, groups and informants in each setting. The study was clearly a very large, complex and is likely to be useful for the countries in it, given their involvement. It will also provide useful information	Reviewer 2 Dr. Julie Leask, University of Sydney, University of Sydney	We appreciate this feedback, and also hope it is directly applicable to countries in need of immunization strengthening.	n/a

	for other countries with weaker immunisation programmes.			
1 3	The findings of the study aren't as clearly connected to the data as qualitative studies often are, and don't usually provide concrete illustrations in a way that makes the themes come alive. This is acknowledged as an inevitability for a study that is so big-picture and the authors point to other papers. However, the authors might consider how to be more specific in certain sections, or to bring more of the quotes and narrative accounts from participants into the results.	Reviewer 2	We acknowledge this feedback and have revised throughout to provide more clarity. We could not include additional quotes - which we agree help illustrate the main text - due to the limited word count. The individual case studies provide more detail and quotes, as this paper is meant to examine the big-picture and impact of this work as a whole.	Edits throughout
1 4	Introduction – This is very well written and addresses some key trends and globally important indicators and their context. There should be a mention of the further decline in coverage after 2019 related to the pandemic, particularly at line 14. As of 2021, coverage had not plateaued, it had experienced a decline in coverage for multiple antigens.	Reviewer 2	We appreciate this feedback. We added a line to address the dip in coverage as a result of the pandemic. However, we want to note that the impact of the pandemic on vaccine delivery was not something we covered in this study as most of the qualitative data was collected prior to the pandemic, and our focus was on historical interventions. Added: The COVID-19 pandemic sent shocks through vaccination programs worldwide, impacting coverage and causing decreases in overall coverage from 2020 on. However, the global impact of the pandemic was not examined as part of this study as a pre-pandemic system was our focus	Introductio n Page 2; Lines 11 1-1114
1 5	Methods. Explain briefly the Exemplars project.	Reviewer 2	Added: This multiple case study analysis was conducted using data from the Exemplars in Vaccine Delivery project within the Exemplars in Global Health program, funded by the Gates Foundation and executed through Gates Ventures. The Exemplars in Vaccine Delivery project involved	Methods Page 4; Lines 15 9-163

			qualitative examination of the national vaccine delivery systems of three low- and middle-income countries – namely Nepal, Senegal, and Zambia – with high childhood routine immunization coverage, compared to their peers.	
1 6	The conceptual model – there is no report of any testing of the model and limited reporting of how it was established. It would be best to call it something other than a model (see later for an alternative suggestion). The APA define models as “a theory, usually including a mechanism for predicting psychological outcomes, intended to explain specific psychological processes.”	Reviewer 2	We provided this model in the Methods section of this manuscript, rather than in the results, because the creation/testing of this model is described in another paper. This model was developed prior to research activities and was used as a guide for the work. To make this clearer, we added: Details regarding the development of this model are available elsewhere [10]. Citation: Published by BMJ Open - Exemplars in vaccine delivery protocol: a case-study-based identification and evaluation of critical factors in achieving high and sustained childhood immunisation coverage in selected low-income and lower-middle-income countries - PubMed (nih.gov)	Methods Page 4; Lines 179-180
1 7	Figure 1 is assumed to be the boxes, but it's not labelled thus in the submission. Assuming that it is the first box, the concepts in it are comprehensive but some are not clearly distinctive. For example, perceived norms/control and attitudes. Or perceived norms and population awareness. What is accommodation/acceptability? While later a table describes these, they relate to the second figure.	Reviewer 2	This model (Figure 1, in the methods section) was developed and described as part of another paper, our protocol paper, which was published in BMJ Open last year. Due to word count restrictions we have opted to cite the paper in lieu of additional details. Citation: Exemplars in vaccine delivery protocol: a case-study-based identification and evaluation of critical factors in achieving high and sustained childhood immunisation coverage in selected low-income and lower-	n/a

			middle-income countries - PubMed (nih.gov)	
18	To reduce cognitive load for readers, acronyms should be minimised (and removed in the abstract). Rather than KIIs and FGDs, is it possible to call these focus groups and interviews? The authors could begin by explaining they undertook key informant interviews, thereafter calling them interviews, and focus groups, thereafter calling them groups. Focus groups are by their very nature discussions – making this word redundant. While I realise these terms are common in global health, it is at least worth considering.	Reviewer 2	Thank you for the feedback. However, in our experience these acronyms are well-known and easy to comprehend.	n/a
19	Study sites – while the authors describe a method for selecting sites elsewhere, this article should at a minimum explain briefly the rationale in the methods.	Reviewer 2	Addressed. Added: Nepal, Senegal, and Zambia were selected as exemplary countries in terms of their routine immunization coverage due to relatively high DTP1 and DTP3 coverage during the timeframe assessed in this study.	Methods section 2.1. Page 4; Lines 190-192
20	The qualitative analysis is described thoroughly. It was theory informed but which theory/theories were used? Was it the conceptual model or other theories? If it was the model, that is not a theory. The authors should state why interviews were used with key informants and focus groups were used with caregivers and community workers. It's a reasonable approach, but each method has unique qualities and should be justified. For the qualitative analysis, who did the coding, who analysed the codes, and how were other team members involved?	Reviewer 2	Our codebook utilized a variety of resources, including the CFIR and CICI frameworks (mentioned in the Methods; in line 181-182); and the conceptual model presented early on in the Methods section. We do not feel that the distinction between a model and a theory is relevant here. Regarding our coding process, we agree this should be included and have added a few details to section 2.2. of the methods: Coding was completed by researchers at Emory, some of whom also developed and revised the codebook throughout. Coding meetings were held frequently to discuss code definitions and to discuss	Methods Section 2.2. Page 5; Lines 277-230

			emerging themes. Researchers from in-country teams were involved through additional meetings to discuss data and ground-truth findings.	
2 1	Section 2.3 – what are catalytic improvements? Also, the quantitative analyses were published elsewhere but no reference is given.	Reviewer 2	Edited to “significant improvements”, although “catalytic improvements” was the wording used in other papers. Thank you for catching this oversight! We have added the citations. ADDED CITATIONS:  • Association of childhood vaccination with family planning, healthcare access, and women education: analysis of Nepal, Senegal and Zambia medRxiv • Factors associated with vaccine coverage improvements in Senegal medRxiv 	Methods Section 2.3. Page 6; Line 265 Citations added to page 6; Line 269
2 2	Findings were disseminated to experts and governments which is said to “ground truth” the results. It would be useful to hear about the mechanisms of feedback which enabled this ground truthing to occur. Presumably, based on the reflexivity statement, there was a way for countries to reflect back and refine the interpretations? Furthermore, it would be interesting to see how this affected the results, for example when the author’s emerging analysis was revised due to the feedback. Could this be mentioned in the reflexivity statement?	Reviewer 2	Addressed by adding the following paragraph to section 2.5. in the methods: The research team at Emory University co-hosted regular meetings with the in-country research partners, experts in a variety of relevant fields, government officials, and other advisors to ask for feedback and contextualization. Researchers would present on methods, findings, and implications following an engaging discussion with advisors that would lead to improvements in our data analysis processes or presentation of findings. Revisions would be distributed through follow-up email conversations and subsequent phone calls with select advisors.	Methods Section 2.5. Page 7; Lines 295 - 300
2 3	Twice it is said the study was hypothesis generating and not intervening. This is unneeded repetition.	Reviewer 2	Addressed by removing the repeated language.	Methods Section 2.6

				Page 7; Line 304
2 4	The authors revise figure 1 because of the over-riding importance of cultural, historical and statutory factors, but the new framework doesn't include these named factors. Openness to a new framework (fig 2) is a sign of good qualitative data analysis. Definitions are given for key domains, which is useful, but that makes figure 1 even more distracting. A suggestion for the authors to consider: in the introduction, state that certain concepts were used to structure what was explored in the data collection, and list these and their sources, but leave figure 2 to be a "conceptual framework" and revise this to include the cultural, historical and statutory factors.	Reviewer 2	Although we appreciate this feedback, we do not believe that any changes to our revised framework (Figure 2) are necessary. The contextual factors encompass the revised framework (Figure 2) via the box labeled "Overarching Domain: Context". Instead of editing the Figure, we have opted to describe this in more detail in the text for clarity: Our empirical data from Nepal, Senegal, and Zambia revealed that the critical success factors could be best understood by detailing understanding and considering the cultural, historical, and statutory context in which the interventions were delivered, which is highlighted in our revised framework as the "Overarching Domain: Context" encompasses all components of the vaccine delivery system. Details about the cultural, historical, and statutory factors that each country's general "context" was composed of are outlined in the individual case study papers [15, 16, 17].	Results Section 3.1. Page 8; Lines 328-334
2 5	In table 2, the definition of community access isn't sufficiently clear and relies heavily on jargon. This is the same with describing an "integrated and horizontal" approach in Health Systems Strengthening. Use plainer language to increase access to the meaning for readers.	Reviewer 2	Addressed: The ability (or inability) to successfully carry out the transaction of vaccine utilization – for example, a community member's ability to access vaccines should they decide to get vaccinated - i.e., barriers and facilitators between Intent and Readiness.	Results – Table 2 Page 9
2 6	As the paper repeatedly highlights, each country's context (eg., history and governance structures) is unique and this is important background. Is it possible for authors to briefly describe each country to give readers context. The results need to	Reviewer 2	We talk more specifically about each country's cultural, historical, statutory, etc. contextual factors in the country-specific individual case study papers referenced throughout. The purpose of this paper is to explore the similarities between the three countries; we hope readers will refer to the	Page 8; lines 332 - 334

	better explain how contexts relate to each of the main themes as these connections aren't clearly made, despite the importance of context.		individual case study papers for more information if they are interested. Added for clarity: Details about the cultural, historical, and statutory factors that each country's general "context" was composed of are outlined in the individual case study papers [15, 16, 17].	
2 7	The results are comprehensive and have face validity. It's a pity to divorce most of the illustrative quotes from the description of the findings since that gives them more authenticity. I prefer to see quotes in context, not in other tables.	Reviewer 2	We agree with this comment. Unfortunately, as this paper has a word count limit, we were not able to include as many quotes in the main text as we would like.	n/a
2 8	The results don't speak much about the views from the focus groups.	Reviewer 2	This was a choice we made due to the limited word count. It was not possible to fit all the information into one paper, which is why we reference the individual case studies. However, we do speak to the views from FGDs in these sections: - Community engagement - Adaptive capacity - Intent to vaccinate - Community access	n/a
2 9	Table 3 – change colours to greyscale for printed versions as blue and green don't differentiate.	Reviewer 2	If this were to be published, we would prefer color for the online version; and greyscale for a printed version.	n/a
3 0	Discussion. This is a key sentence, but its meaning isn't clear enough: "Describing how national governance structures operate with the consideration of contextual factors was essential to our understanding of how global guidelines should be adapted to reflect local priorities and challenges."	Reviewer 2	Edited this sentence for clarity: In order to understand how global guidelines should be adapted to reflect local priorities and challenges, it was important for us to describe how national governance structures operate with the consideration of contextual factors.	Discussion Page 13; lines 497 - 498

3 1	Line 35 – at this point, there is a puzzling absence of any discussion of the impact of the pandemic. In Senegal, this would have been discussed in interviews given the timeframe for data collection. could consider further reflecting on this characteristic in the discussion, which has been found elsewhere to create resilience against other risks.	Reviewer 2	As this study employs a historical perspective, we did not ask participants to focus on the impacts of the pandemic, and we did not include these findings in the analysis utilized for this paper. The objective was to explore success factors from these three countries and assess similarities. The pandemic presented unique challenges for us, but we did our best to tailor the data collection tools to pre-2019 activities as per our study aims. Another paper was published by our research team (led by in-country PIs) to specifically look into the impact of the pandemic. This is cited in the conclusion: Dixit, S.M., et al., Addressing disruptions in childhood routine immunisation services during the COVID-19 pandemic: perspectives from Nepal, Senegal and Liberia. BMJ Global Health, 2021. 6(7): p. e005031.	n/a
3 2	In the supplementary materials, the search strategy is about falls. Please clarify.	Reviewer 2	We have updated the supplemental checklist; please see attached.	n/a

VERSION 2 – REVIEW

REVIEWER	Leask, Julie University of Sydney, School of Public Health
REVIEW RETURNED	12-Jun-2023

GENERAL COMMENTS	Some of the comments have been responded to, but others not sufficiently in my view and sometimes not at all. There is an over-reliance on referring to previous papers to provide essential detail for this paper. This paper needs to stand alone. Each country should be briefly described for this paper, even if it's a basic two or three sentences. There remains terminological confusion about models, theories and frameworks. Conceptual model and Framework are used interchangeably without consistency of application.
---

	In relation to theory informed qual analysis, using CFIR and CICI to structure the interview guides does not make it a theory informed analysis. In any case, these are frameworks. The authors need to specify with actual theories they used in the analysis, if any or simply say the analysis was informed by the domains in figure 1. The addition of detail about analysis is useful. Given that reviewer 1 had comments about context too, I reiterate the recommendation to add the words about context back into figure 2 (which has not been provided in the revised version). The authors have not addressed this: “The authors should state why interviews were used with key informants and focus groups were used with caregivers and community workers. It’s a reasonable approach, but each method has unique qualities and should be justified.” Re colours of the figures – I recommend the colours used are sufficiently distinctive. A person may print an online version and be unable to readily distinguish the colours, as I was. Abstract: Give number interviewed in results not methods. This part of the sentence in abstract is problematic and needs to be revised. “Although vaccine coverage improvements across Africa and South Asia have remained relatively stagnant and below global targets..” Improvements denote increase, not stagnation and as a statement in the present tense, it’s not accurate as coverage has dropped, not stagnated, regardless of what period this study refers to.
--	--

VERSION 2 – AUTHOR RESPONSE

There is an over-reliance on referring to previous papers to provide essential detail for this paper. This paper needs to stand alone. Each country should be briefly described for this paper, even if it’s a basic two or three sentences.	Reviewer 2 Dr. Julie Leask, University of Sydney	Thank you for the feedback. We have added brief descriptions of the results from each individual case study to Section 3.2. - Operationalization of successful vaccine delivery programs. Unfortunately, we are unable to provide much more information in-text due to word count limits. We have re-worked our methods section to hopefully provide additional information and make it clear that interested individuals can look further for additional details. We hope this is satisfactory. Added: “In Nepal, public codification of health (and therefore, vaccines) as a human right ensured the stability of	304-317
---	--	---	----------------

		vaccination programming, increased morale among health workers, and helped to generate demand for vaccines. Female Community Health Volunteers (FCHVs) and other community health workers lead a powerful community-based approach to demand generation, health education, and vaccine delivery, which was supported by cultural values including collective responsibility and community ownership of vaccine coverage.[16] In Senegal, strong political will supported the prioritization of vaccination programming, including urgent allocation of resources and cold chain improvements. Long-term partnerships between government agencies and external partners allowed for innovation, capacity building, and efficiency within the health system. Lastly, improved surveillance efforts led to timely and evidence-based decision making.[17] In Zambia, effective collaboration was essential to successful vaccine delivery. The Inter-Agency Coordinating Committee led long-term collaboration efforts at the national level and supported evidence-based programming through multiple technical working groups. At the local level, Neighborhood Health Committees spearheaded community-driven strategies via community action planning and strengthened the link between communities and health facilities. [15]"	
There remains terminological confusion about models, theories and frameworks. Conceptual model and Framework are used interchangeably without consistency of application.	Reviewer 2	Thank you for this feedback; we have done a review of the terms to address any confusion. We reviewed the definitions of theory, model, and framework from the book "Implementation Science 3.0" by Per Nilsen, addressed in the chapter "Making Sense of Implementation Theories, Models, and Frameworks" (https://link.springer.com/chapter/10.1007/978-3-030-03874-8_3). According to this source, we found:  • A theory is a set of "principles or statements designed to structure our observation, understanding and explanation of the world" • A model "involves a deliberate simplification of a phenomenon or a specific aspect of a phenomenon" • A framework denotes "structure, overview, outline, system, or plan consisting of various descriptive categories" 	Throughout the paper

		Changes made: After this review, we edited the manuscript to use the term “framework” when referring to Figure 1 and Figure 2.	
In relation to theory informed qual analysis, using CFIR and CICI to structure the interview guides does not make it a theory informed analysis. In any case, these are frameworks. The authors need to specify with actual theories they used in the analysis, if any or simply say the analysis was informed by the domains in figure 1.	Reviewer 2	Thank you for this clarification. We have updated our language to reflect that the domains from our initial framework informed the analysis. Edited: “We applied thematic analysis of the transcripts to identify critical success factors. Analysis was informed by the domains outlined in Figure 1.”	179-180
Given that reviewer 1 had comments about context too, I reiterate the recommendation to add the words about context back into figure 2 (which has not been provided in the revised version).	Reviewer 2	We have edited the figure to highlight the importance of context. All previous iterations of the figure included context as an overarching domain – for this version, we increased the font size and added a few key words to make sure that it is notable. Please let us know if this sufficiently addresses your feedback.	n/a
The authors have not addressed this: “The authors should state why interviews were used with key informants and focus groups were used with caregivers and community workers. It’s a reasonable approach, but each method has unique qualities and should be justified.”	Reviewer 2	We addressed this comment by adding the following information to the Methods under Section 2.2. Qualitative data collection and analysis. Text added: “KIs were held with government officials in order to understand their current and historical perspectives on the national vaccination program based on their official positions. Given inherent power dynamics, we decided that FGDs would have been inappropriate. At the community level, we conducted FGDs because we hoped that group conversations would allow for interactive discourse on the key themes.”	161-165

Re colours of the figures – I recommend the colours used are sufficiently distinctive. A person may print an online version and be unable to readily distinguish the colours, as I was.	Reviewer 2	Thank you for your feedback. However, the figures are consistent with other papers published as a result of this project, and they are also consistent with the figures posted on the Vaccine Exemplars website. Other reviewers have not commented on the colors, and we think it is important to keep them consistent across papers and platforms. No changes made.	n/a
Abstract: Give number interviewed in results not methods.	Reviewer 2	We have updated our abstract as directed by the Editor, and therefore have included this information in the “participants” section, For additional information please see: https://bmjopen.bmj.com/pages/authors/#research Participants section: “We conducted 66 key informant interviews and 22 focus group discussions with a total of 678 participants. Participants were recruited from all levels, including government officials, health facility staff, frontline workers, community health workers, and parents. Participants were recruited from both urban and rural districts in Nepal, Senegal, and Zambia.”	30-33
This part of the sentence in abstract is problematic and needs to be revised. “Although vaccine coverage improvements across Africa and South Asia have remained relatively stagnant and below global targets..” Improvements denote increase, not stagnation and as a statement in the present tense, it’s not accurate as coverage has dropped, not stagnated,	Reviewer 2	We have addressed this by removing the phrase “relatively stagnant and” – this now reads: “Although vaccine coverage improvements across Africa and South Asia have remained below global targets...”	21

regardless of what period this study refers to.			
---	--	--	--